# Use of a cooling pack to reduce subcutaneous vaccine injection pain in children aged 3–6 years: A single-blind, randomized, parallel-group multicenter study

**Ikuo Okafuji[1]\*, Ryuta Washio[2], Takao Okafuji[3], Yu Tanaka[1,4], Tatsuo Kagimura[5]**

1 Department of Pediatrics, Kobe City Medical Center General Hospital, Kobe, Hyogo, Japan, 2 Washio Child Clinic, Kobe, Hyogo, Japan, 3 Okafuji Pediatric Clinic, Himeji, Hyogo, Japan, 4 Department of Pediatrics, Kobe University Graduate School of Medicine, Kobe, Hyogo, Japan, 5 Translational Research Center for Medical Innovation, Foundation for Biomedical Research and Innovation at Kobe, Kobe, Hyogo, Japan

\* okafuji@kcho.jp

## Abstract

### Background

There are many evidence-based treatments available for pain-relief during subcutaneous vaccine injection in children. However, these methods are commonly labor-intensive and not routinely applied in clinical settings.

### Objectives

To evaluate the safety and pain-relieving effect of local cooling during subcutaneous vaccine injection in young children.

### Method

This single-blind, randomized, parallel-group multicenter study was conducted at two pediatric clinics in Hyogo Prefecture, which included preschool children aged 3–6 years scheduled for vaccination against Japanese encephalitis or influenza virus. Participants were randomly assigned to either the cooled group (local cooling applied using a cooling pack before vaccination) or the non-cooled group (provided with a room temperature cooling pack). Randomization was performed using a computer-generated block method. The primary endpoint was infant pain, measured using the FLACC scale (Facial expression, Leg movement, Activity, Crying, Consolability), rated by a third-party reviewing videos of the vaccination process.

### Results

A total of 60 children were randomized into the cooling (n=30) and non-cooling (n=30) groups, with all participants completing the study. Fifty-four participants received the Japanese encephalitis vaccine, and six received the influenza vaccine. Demographic data,

**Data availability statement:** The data supporting the findings of this study have been deposited in the Dryad, an open data publishing platform and community committed to the open availability and routine re-use of all research data, and are available at https://doi.org/10.5061/dryad.4xgxd25kw.

**Funding:** Funding sources: 1. Katakami Foundation for Clinical Research. 2. Mie Chemical Industry Co., Ltd. 3. Tanac Co., Ltd . Initials of the authors who received each award: 1. IO 2. IO 3. IO Grant numbers awarded to each author: NA URL of each funder website: 1. https://chuo.kcho.jp/media/chuo/internal/ccir/pdfs/researcher/subsidy/fund_katakami_summary_r3.pdf 2. https://mie-kogyo.co.jp 3. https://global.k-tanac.co.jp Did the sponsors or funders play any role in the study design, data collection and analysis, decision to publish, or preparation of the manuscript? Answer: No, Tanac co., ltd and Mie chemical industry co., ltd provided some materials used in this study. Tanac co., ltd also covered the costs of the specific clinical research review.

**Competing interests:** The authors have declared that no competing interests exist.

including age, sex, and vaccine type, did not differ significantly between the two groups. The median FLACC score in the cooled group was significantly lower (1 [IQR 0–1.25]) compared to the non-cooled group (2.5 [IQR 1–6]) (P = 0.011). No adverse effects related to cooling were observed.

## Conclusion

Local cooling during subcutaneous vaccine administration is a safe and effective method to reduce pain in children aged 3-6 years. This method can be easily implemented in routine vaccinations to improve patient comfort.

## Trial registration

Japan Registry of Clinical Trials, jRCTs052200149, Mar 09, 2021, https://jrct.niph.go.jp/en-latest-detail/jRCTs052200149.

## Introduction

Childhood anxiety and fear of vaccine pain can contribute to a lifelong aversion to vaccination [1,2]. This aversion may persist into adulthood, impacting compliance with necessary vaccinations, thereby compromising overall public health [3]. Consequently, vaccination noncompliance contributes to outbreaks of vaccine-preventable diseases and undermines the individual and community benefits of vaccination [3,4]. A group of Canadian experts in child health and vaccines developed clinical practice guidelines on pain reduction during vaccination, including pharmacological interventions, such as local anesthetics; physical interventions, such as positioning during vaccine injection; and procedural interventions, such as not using suction during vaccination [5]. The Global Advisory Committee on Vaccine Safety of the World Health Organization has since proposed a new concept termed Immunization Stress-Related Responses to prevent, diagnose, and respond to various stress-related reactions to vaccination [6]. This concept refers to specific responses to pain, including posture and talking to the patient during vaccination. In Japan, pain management for needle-related procedures is generally considered to be inadequate. Yamamoto-Hanada et al. combined a pharmacological approach with lidocaine/prilocaine cream and non-pharmacological approaches, such as preparation, education, positioning, and distraction to create a multidisciplinary pain management method that can be implemented in children, and has been reported to be effective in pain relief [7]. A Cochrane review of psychological interventions for needle-related procedure pain and distress in children found that despite low-quality evidence, there are potential benefits of pain or distress reduction supports using these interventions in clinical practice [8]. As such, there are many evidence-based treatments available to reduce pain during vaccination; however, many involve significant time and resource commitments, compromising their feasibility for routine implementation [9–11].

Among the various available non-pharmacological methods, local cooling has gained attention as a rapid and simple approach to reduce pain [11–13]. While vapocoolant sprays have demonstrated efficacy in adults, their effectiveness in children remains limited, with studies indicating that young patients may perceive the cold sensation as unpleasant [9,14,15]. Although systematic studies are required to validate these observations, anecdotal observations in clinical practice have indicated that gel-based cooling packs, which maintain softness even when frozen, represent a less intimidating and potentially more effective alternative for pediatric patients. This feature could potentially facilitate more comfortable and effective pain-relief during vaccination.

To explore this hypothesis, we conducted a single-blind, randomized, parallel-group multicenter study to evaluate the effectiveness of local cooling using a gel-based cooling pack at reducing pain associated with subcutaneous vaccine injections in children aged 3 to 6 years. The primary objective of this study was to assess the pain-relieving effect of local cooling during vaccination, as measured using the Facial Expression, Leg Movement, Activity, Crying, and Consolability (FLACC) scale [16–20], which is widely recognized for its reliable assessment of pain in young children. Additionally, we aimed to evaluate the safety and feasibility of this method in a clinical setting.

## Materials and methods

### Trial design

This randomized, single-blind, placebo-controlled, multicenter trial was conducted across two pediatric clinics in Hyogo, Japan, from May 26, 2021, to Nov 2, 2021. The Research Ethics Committee of Kobe City Medical Center General Hospital granted ethical approval (approval number CRB5190001). The study was registered under the Japanese Clinical Trial Registry number jRCTs052200149. Written informed consent was obtained from all study participants.

### Participants

Participants aged 3 to 6 years scheduled for vaccination against Japanese encephalitis virus or influenza virus were deemed eligible to participate (Fig 1). The exclusion criteria were as follows: (1) excessive crying by the child before obtaining consent (if a child in this age group cries violently, touch alone is sufficient to induce even more violent crying which can prevent correct pain assessment); (2) presence of perceptual dullness attributed to underlying conditions, such as an existing medical ailment; (3) scheduling of two or more vaccinations on the same day, as consent was acquired; and (4) physician's assessment determining the child's unsuitability for study participation, based on considerations of the child's personality traits, including tendencies toward heightened anxiety.

### Vaccines used

For the Japanese encephalitis vaccine, each clinic used the vaccine routinely administered at their respective site. Specifically, the O Clinic administered ENCEVAC, manufactured by KM Biologics Co., Ltd., while the W Clinic used JEBIK V, produced by the Research Institute for Microbial Diseases of Osaka University (BIKEN). Both clinics used the INFLUENZA HA VACCINE "BIKEN" containing thimerosal, also produced by the Research Institute for Microbial Diseases of Osaka University, as the influenza vaccine.

### Randomization and masking

After the childrens' eligibility was verified and written consent was obtained, participants were randomly assigned to the cooling and non-cooling groups. REDCap 11.1.19 (Vanderbilt university, Nashville, TN, USA) electric data capture tool hosted at our hospital was used for computer randomization, stratified by a block size of four. The allocation was concealed from the participants, guardians, and assessors.

### Trial procedures

For topical cooling, we used a cooling pack that retains its soft shape even when frozen (Fig 2A). In this study, a product made for anti-inflammatory and analgesic purposes (PURU CURE Ice Pack®) was modified to be sufficiently small to fit the infant's upper arm (Fig 2B). Moreover, the cooling pack reduced skin surface temperature when applied to the local area

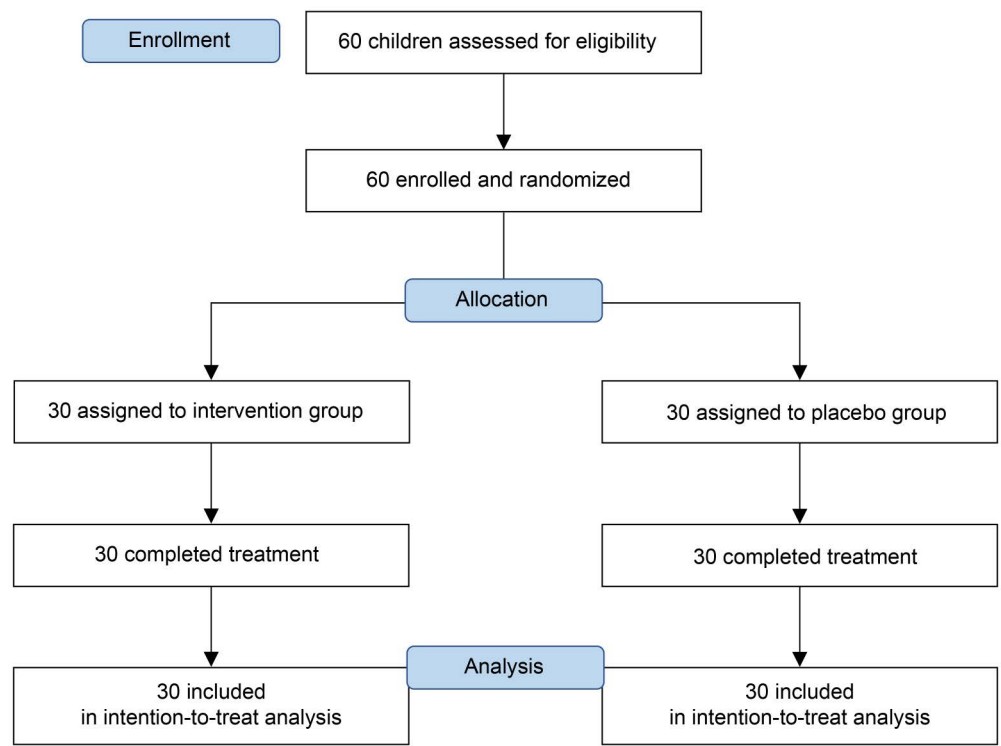

**Fig 1. Enrollment and randomization of participants in the study.** Children aged 3–6 years who presented to the clinic for vaccination against Japanese encephalitis virus or influenza virus, who were not crying when they entered the clinic, were recruited and provided consent for inclusion in the study.

after being cooled in a household freezer before use, but not when applied after being left at room temperature. Cooling packs placed in ice water for 1–3 h after cooling for at least 3 h in a freezer with an internal temperature of -16 to -20 °C were used in the cooling group. Cooling packs left at room temperature (approximately 20 °C) for over 3 h were used for the non-cooled group. Fig 3 depicts the study design and timing of the cooling pack application and data recording.

After written consent was obtained from the guardian on the vaccination day, the participant was enrolled in REDCap and assigned to the study using REDCap by medical staff in the clinic. The participant and his/her guardian entered the vaccination area together, and the participant received the vaccination in the posture of his/her choice. Lidocaine/prilocaine cream was not applied in this study, as it is not routinely used in vaccination settings in our country, including in either of the two participating clinics. A medical staff member applied a cooling pack to the upper arm of the participant for 30–60 seconds, being careful not to reveal the allocation to ensure that the guardian, who would later become the pain evaluator, was blinded to the allocation. Furthermore, within 60 seconds of removing the cooling pack, the physician administered the vaccine to the upper arm of the participant on the side to which the cooling pack had been applied. Vaccination was performed in accordance with the WHO position paper [6] by one and three pediatricians at the O and W clinics, respectively. The time between applying the cooling pack and 10 seconds after vaccination was videotaped so that a third party could later assess the FLACC scale rating. The guardian scored the FLACC scale rating (Table 1) from the time the participant and his/her guardian left the vaccination area after vaccination to the time they left the clinic.

(A) (B)

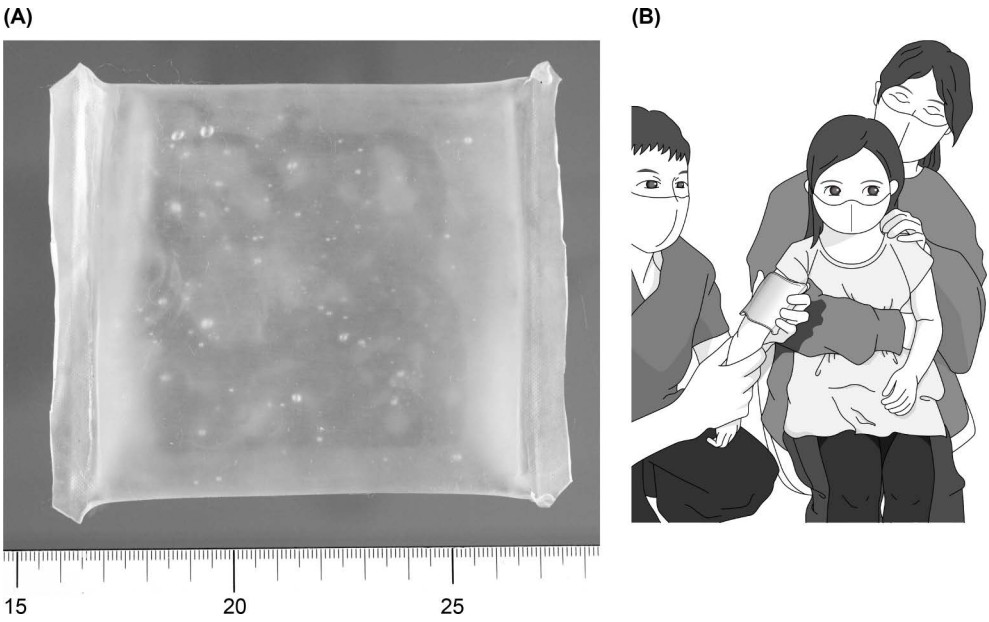

**Fig 2. Scene at the time of local cooling before vaccination.** (a) The cooling pack was sized to appropriately fit on the upper arm of the infant: a scale in centimeters is shown at the bottom to indicate the length. The color of the gel filling the cooling packs was uniformly yellowish-green. (b) While cooling the site to be vaccinated, a cooling pack was applied to the upper arm of the child by medical staff, taking care not to reveal the allocation to the guardian.

## Outcomes

The primary endpoint was procedure-related pain, assessed using the FLACC scale by a third party who later watched the recorded video to determine the condition of the patient at the time of vaccination. The FLACC score is a pain rating scale for affected children which consists of a total score of five sub-items: facial expression, leg movements, activity, crying, and consolability, with the sub-items rated as 0, 1, or 2 according to the rating criteria presented in Table 1 [16–19]. Because previously reported FLACC categories and total scores showed high agreement among observers [17], and the preliminary results in this study were similar, the third-party evaluation was performed by only one specific person in this study. The secondary outcome was the FLACC scale rating provided by the guardians.

## Statistical analysis

Assuming an effect size of 0.9 on the FLACC scale as the primary endpoint, the number of patients required to detect a difference with 80% power at a two-sided 5% significance level was calculated using an unpaired t-test, yielding a sample size of 27 per group. The sample size calculation was based on methodologies outlined in previous studies [21,22]. Although the primary analysis of this study was designed to compare groups using the Wilcoxon rank-sum test for the FLACC score, we were unable to identify the standardized effect size for the U statistic in the literature necessary to calculate the sample size using the Wilcoxon test. Therefore, we used the unpaired t-test for sample size calculation, which may offer higher power for the given assumptions. Considering the lower power of the nonparametric test planned for the primary analysis and potential dropouts due to inadequate videotaping, the final sample size was set at 30 patients per group.

In accordance with the intention-to-treat (ITT) principle, the efficacy analysis set was defined as the ITT population, including all randomized patients. Data are expressed as

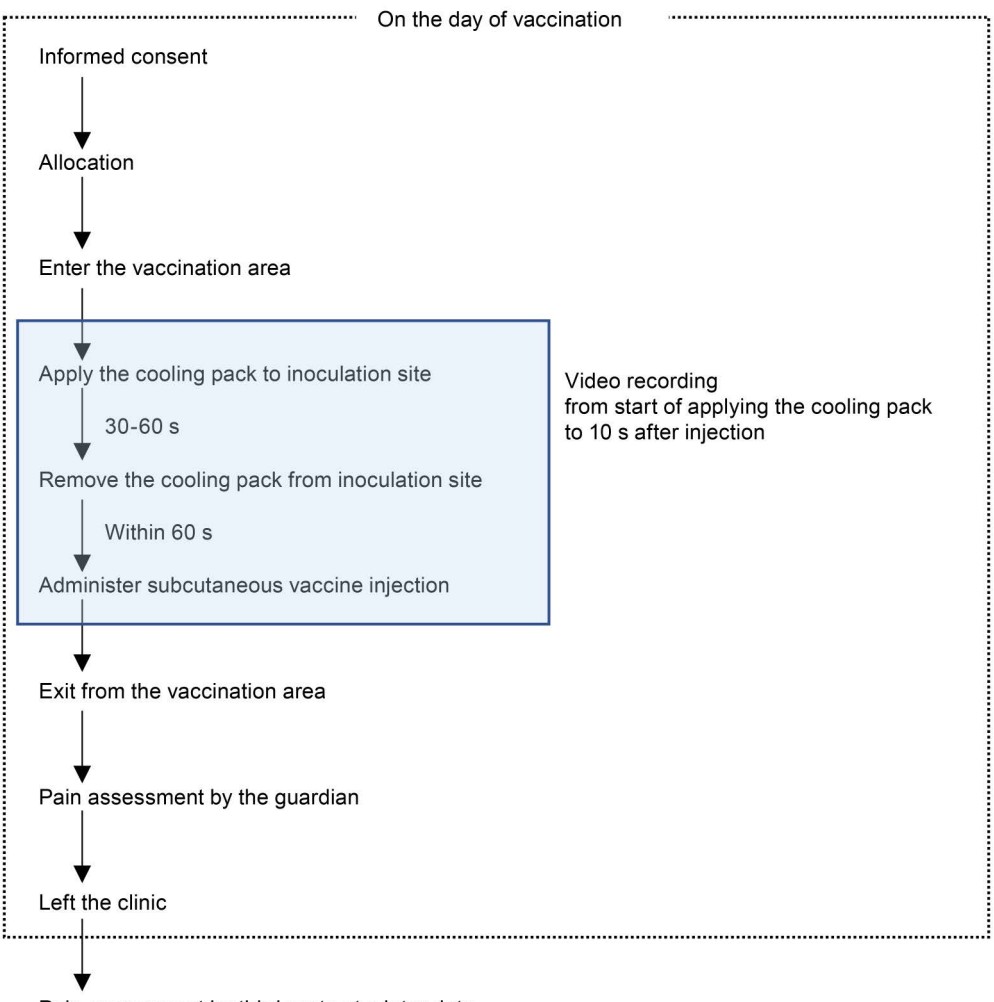

**Fig 3. Study flowchart.** Participants were assigned to the cooled or non-cooled group after receiving an explanation on the day of vaccination, and indicating their willingness to consent in writing. Participants were then exposed to a cooling pack, either cooled or non-cooled, for 30–60 s, after which they were vaccinated within 1 min, which was videotaped. The guardians used the time between the end of the vaccination and leaving the hospital to assess the pain experienced by the child during the vaccination. Further, a third party later viewed the video and rated the pain of the child during vaccination.

the medians and interquartile ranges (IQRs) for continuous variables, and frequencies and percentages for categorical variables. FLACC scores were compared between groups using the Wilcoxon rank-sum test, while inter-rater associations were examined using scatter plots and Spearman's rank correlation coefficients. Imputation of missing values was not performed for either the primary or secondary end point. Baseline characteristics between groups were compared using the Wilcoxon rank-sum test, χ2 or Fisher's exact test, depending on the characteristics of the variables. The significance level was set to P < 0.05 (2-tailed). Statistical analysis was performed using JMP14.0 (SAS Institute, Cary, NC, USA).

## Results

Sixty children were evaluated for study eligibility and subsequently randomly assigned to one of the groups during the study period between May 26, 2021, and November 2, 2021.

**Table 1. Overview of the FLACC behavioral pain assessment scale.**

| | 0 | 1 | 2 |
|---|---|---|---|
| Face | No particular expression or smiling | Occasional grimace or frown; withdrawn, disinterested | Frequent to constant frown, clenched jaw, quivering chin |
| Legs | Normal position or relaxed | Uneasy, restless, tense | Kicking or legs drawn up |
| Activity | Lying quietly, normal position, moving easily | Squirming, shifting back and forth, tense | Arched, rigid, or jerking |
| Cry | No crying | Moaning or whimpering, occasional complaint | Crying steadily, screams or sobs; frequent complaints |
| Consolability | Content, relaxed | Reassured by occasional touching, hugging, or being talked to; distractible | Difficult to console or comfort |

The FLACC scale assesses pain in children, yielding scores from 0 to 10 based on five categories: facial expression, leg movement, activity, crying, and consolability.

FLACC, facial expression, leg movement, activity, crying, and consolability.

Therefore, data from 60 children were available for analysis (Fig 1). Table 2 shows the baseline characteristics for both groups. The median age was 3 years; 42 children were 3 years old, 8 were 4 years old, 7 were 5 years old, and 3 were 6 years old. There were 29 boys, 14 of whom were in the intervention group and 15 in the placebo group. The remaining 31 participants were girls, with 16 in the intervention group and 15 in the placebo group. Fifty-four children were administered the Japanese encephalitis virus vaccine, of whom 27 each were randomized to the intervention and placebo groups. Six children were immunized against the influenza virus: three in the intervention group and three in the placebo group.

We identified a significant difference between the cooling group and non-cooling groups (P = 0.011) in the primary study outcome. The median FLACC score assessed by a third party was 1 [IQR 0–1.25] in the cooling group and 2.5 [IQR 1–6] in the non-cooling group (Table 3). The secondary endpoint, the median FLACC score assessed by the guardian, was 0 [IQR 0–1.25] in the cooling group and 1 [IQR 0–4] in the non-cooling group, which was also significantly different (P = 0.042). There were no side effects in either group.

The FLACC scale evaluated by the guardian immediately following vaccination was lower than that evaluated by the third party after vaccination by watching a video of the vaccination; however, there was a correlation between these variables (Fig 4). Among the five components of the FLACC scale, the scores for "Cry" and "Activity" were positively correlated between the third-party and guardian evaluations (Spearman's rank correlation coefficients: 0.64 and 0.51, respectively), while the scores of "Face," "Leg," and "Consolability" were weakly correlated between the third-party and guardian evaluations (correlation coefficients: 0.44, 0.47, and 0.37, respectively) (Table 4).

## Discussion

This is the first study to reveal that applying local cooling prior to vaccination can alleviate pain associated with vaccination in young children aged 3–6 years. Childhood anxiety and fear of pain can trigger a lifelong phobia of needles and vaccinations, and is an important factor influencing healthcare avoidance in general [3]. The pain relief method implemented in this study is simple and convenient. Therefore, it could be used without burdening clinical practice in acute and urgent procedures and planned procedures such as vaccination.

The primary difficulty in assessing pain in children is that subjective assessments used in adults, such as VAS and numerical rating scales, cannot be used [14]. Hence, several objective pain assessment methods for use in children have been developed. The FLACC scale and Children's Hospital of Eastern Ontario Pain Scale are both reliable and valid objective

**Table 2. Baseline characteristics.**

| | Intervention Group (n = 30) | Control Group (n = 30) | P-value |
|---|---|---|---|
| Age, median years (range) | 3 (3–6) | 3 (3–6) | 0.32 |
| Sex: | | | 0.80 |
| Male, n (%) | 14 (47) | 15 (50) | |
| Female, n (%) | 16 (53) | 15 (50) | |
| Vaccine against: | | | 1.00 |
| Japanese encephalitis virus, n (%) | 27 (90) | 27 (90) | |
| Influenza virus, n (%) | 3 (10) | 3 (10) | |
| Facility: | | | 1.00 |
| Clinic O, n (%) | 11 (37) | 11 (37) | |
| Clinic W, n (%) | 19 (63) | 19 (63) | |

P-values were calculated using the Wilcoxon rank-sum test for continuous variables (age) and the Fisher's exact test for categorical variables (sex, vaccine type, and facility).

n: number of participants.

**Table 3. Comparison of median FLACC Scale by evaluator.**

| Evaluator | Intervention group (n = 30) Median [IQR] | Control group (n = 30) Median [IQR] | P-value |
|---|---|---|---|
| The third party | 1 [0–1.25] | 2.5 [1–6] | 0.011 |
| The guardian | 0 [0–1.25] | 1 [0–4] | 0.042 |

FLACC, facial expression, leg movement, activity, crying, and consolability.

IQR, interquartile range.

P-values were calculated using the Wilcoxon rank-sum test.

pain assessment methods for use in toddlers and older children [23,24]. The FLACC scale, for which a validated Japanese version has been constructed, was selected as the assessment method for this study [17]. A third party who watched a video recording of the vaccination scene evaluated the FLACC scale. As such, the FLACC scale could be evaluated in all enrolled cases without any problems, allowing the study to be performed.

In the present study, the FLACC scale was evaluated by either the third party watching the video, or the guardian immediately after vaccination. The former served as the primary end-point and the latter as the secondary. In this study, the cooling group experienced better pain relief than the non-cooling group. The FLACC score of the guardian evaluation was lower than that of the third-party evaluation, and the guardian evaluation was rated lower than the third-party evaluation for the items "Face," "Leg," and "Consolability" on the FLACC scale. The FLACC scale appeared to be the best evaluated by the third-party video viewing method in a clinical research setting [18,20].

Vapocoolant sprays are commonly used for local cooling, and have shown efficacy for pain relief [9]. However, the analgesic effect in infants is limited [14,25]. The sound generated during the use of vapocoolant spray in children is thought to increase the levels of anxiety and stress before the procedure. Hence, a gel-based cooling pack that does not harden even when frozen was used as a local coolant in this study. The tactile feel of the gel was appealing to the children, and unlike the vapocoolant spray, this cooling pack can reduce the anxiety and stress before the procedure, which may also have helped in assessing the local cooling effect.

Both local cooling and topical anesthetics methods offer distinct advantages and limitations as pain relief methods. The utility of topical anesthetics, such as EMLA cream, are well-supported by clinical evidence [7,13,14,26], and are often covered by insurance, minimizing

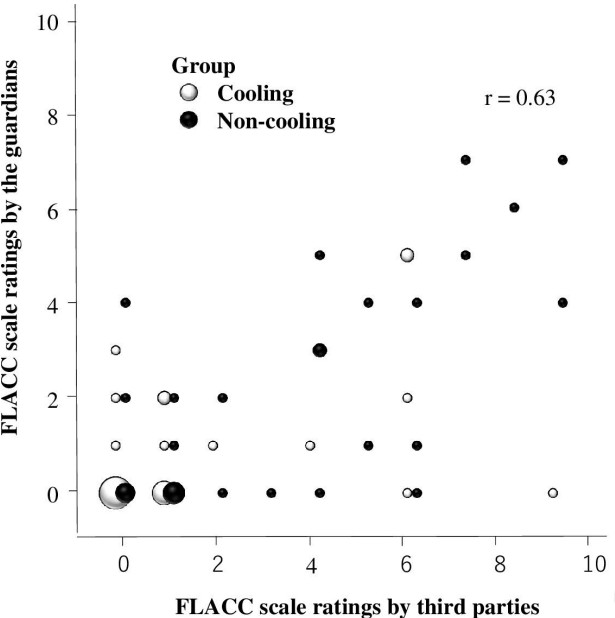

**Fig 4. Scatter plots showing the relationship between third-party ratings (X-axis, independent variable) and guardian ratings (Y-axis, dependent variable) for the FLACC scale.** The dot size reflects the number of participants who received the same rating (e.g., 15 participants received a rating of 0 on both axes). FLACC, facial expression, leg movement, activity, crying, and consolability.

**Table 4. Comparison between observed behaviors at the time of vaccine injection using the Face, Legs, Activity, Cry, and Consolability (FLACC) scale, as evaluated by a third party and the guardian.**

| Component of the FLACC scale | Median value of FLACC component [IQR] | | Spearman's rank correlation coefficient |
|---|---|---|---|
| | The third party | The guardian | |
| Face | 1 [0, 1] | 0 [0, 1] | 0.44 |
| Legs | 0 [0, 1] | 0 [0, 1] | 0.47 |
| Activity | 0 [0, 0] | 0 [0, 0] | 0.51 |
| Cry | 0 [0, 1] | 0 [0, 1] | 0.64 |
| Consolability | 0 [0, 1] | 0 [0, 1] | 0.37 |

FLACC, facial expression, leg movement, activity, crying, and consolability.

IQR, interquartile range.

The Spearman's rank correlation coefficient was calculated to assess the correlation between the evaluations made by the third party and the guardian for each component of the FLACC scale.

patient costs. However, they require 30 minutes to 1 hour for effect and, as a pharmacological intervention, carry the risk of side effects. In contrast, local cooling offers rapid pain relief within 30 to 60 seconds and has minimal side effects, making it a more practical option for pediatric use. Further, cooling packs are reusable, which may reduce costs over time, although they do require upfront investment and preparation by hospitals. If purchased individually by families, the cost is approximately 3,200 yen per unit. While cost-effectiveness in this context has not been rigorously investigated, these factors may influence their broader implementation. Both methods can provide effective pain management; however, the choice should depend on clinical circumstances and patient needs.

This study was subject to some limitations. Firstly, it was conducted at only two institutions, potentially limiting the generalizability of the findings. Nevertheless, all of the enrolled patients successfully completed the study, without any missing data. Secondly, the single-blind design posed a risk of execution bias, as physicians may guess the allocation based on the temperature sensation induced by the cooling pack. Implementing a double-blind design would have been ideal, but was challenging due to the study's nature. Thirdly, the study specifically targeted children receiving Japanese encephalitis or influenza vaccines, which may not elicit significant discomfort compared to other vaccines, such as pneumococcal vaccines. Supply issues with the Japanese encephalitis vaccine have also affected case collection. While baseline anxiety was not explicitly measured, randomization minimized the influence of baseline differences, including anxiety levels, between groups. However, measuring baseline anxiety in future studies could provide valuable insights into its role in pain perception and intervention efficacy. Another limitation was the use of the FLACC scale, which, although widely validated, may not fully capture the influence of psychological factors, such as fear and anxiety, on pain expression in young children. Furthermore, the lack of validated Japanese tools to assess anxiety and fear in preschool-aged children limited our ability to comprehensively evaluate these psychological factors. Finally, there was a gap of 3 years between performance of the investigation and article submission due to challenges such as clinical workload, administrative hurdles, and resource constraints during the COVID-19 pandemic. Despite these delays, these findings remain relevant for alleviating injection-related pain in children. Moreover, this study focused on children aged 3 to 6 years. Given the developmental differences across age groups, further studies are required to assess the effectiveness of cooling packs in other age ranges, including infants and adolescents.

## Conclusions

This study revealed that local cooling is a simple and easy method to alleviate pain associated with vaccination in young children. Adequate pain relief during childhood vaccination may prevent not only vaccine avoidance in children, but also needle fear and vaccine hesitancy, which have arisen as important problems in recent years and are linked to the refusal of medical care in adulthood.

## Supporting information

**S1 File. CONSORT checklist.**
(PDF)

**S2 File. Trial protocol.**
(DOCX)

**S3 File. Statistical analysis plan.**
(DOCX)

## Acknowledgments

We would like to thank Dr. Shigenori Kusuki, Dr. Urara Koudera, and Dr. Miharu Okafuji for recruiting patients and conducting the clinical trial, Ms. Rio Yasaka for her evaluation of the FLACC scale, and the staff at Washio Child Clinic and Okafuji Pediatric Clinic for their contribution to the smooth implementation of our study. We would also like to express our deepest appreciation to Ms. Aya Ako, Ms. Utako Shirono, Ms. Asuka Kiku, Ms. Yoko Miyake, Chisato Miyakoshi, PhD, Chiaki Sakai, PhD (Center for Clinical Research and Innovation)

and Daisuke Yamashita, PhD (Department of Pathology) of Kobe City Medical Center General Hospital, and Takeshi Morimoto, PhD (Department of Clinical Epidemiology, Hyogo Medical University) for their assistance. Finally, we would like to thank Honyaku Center Inc. for English language editing.

## Author contributions

**Conceptualization:** Ikuo Okafuji, Tatsuo Kagimura.

**Data curation:** Yu Tanaka, Tatsuo Kagimura.

**Formal analysis:** Ryuta Washio, Takao Okafuji.

**Investigation:** Ryuta Washio, Takao Okafuji.

**Supervision:** Tatsuo Kagimura.

**Writing – original draft:** Ikuo Okafuji.

**Writing – review & editing:** Ikuo Okafuji, Ryuta Washio, Takao Okafuji, Yu Tanaka, Tatsuo Kagimura.

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
