## [Decision Letter · Decision Letter 0]

13 Sep 2024

PONE-D-24-27274Use of a cooling pack to reduce subcutaneous vaccine injection pain in children : a two-group, single-blind, randomized, parallel-group multicenter studyPLOS ONE

Dear Dr. Okafuji,

Thank you for submitting your manuscript to PLOS ONE. After careful consideration, we feel that it has merit but does not fully meet PLOS ONE’s publication criteria as it currently stands. Therefore, we invite you to submit a revised version of the manuscript that addresses the points raised during the review process.

**Dear Authors,**

Thank you for your submission. Please revise the manuscript thoroughly, addressing each of the reviewers' comments point by point in detail, ensuring that all concerns and suggestions are adequately resolved. If you are unable to revise the paper according to the reviewers' expectations, I would be open to considering it as a letter for publication.

Best,

We look forward to receiving your revised manuscript.

Kind regards,

Academic Editor

PLOS ONE

Journal Requirements: When submitting your revision, we need you to address these additional requirements. 1. Please ensure that your manuscript meets PLOS ONE's style requirements, including those for file naming. The PLOS ONE style templates can be found at https://journals.plos.org/plosone/s/file?id=wjVg/PLOSOne_formatting_sample_main_body.pdf and https://journals.plos.org/plosone/s/file?id=ba62/PLOSOne_formatting_sample_title_authors_affiliations.pdf 2. In the online submission form, you indicated that "Deidentified individual participant data (including data dictionaries) will be made available, in addition to study protocols, the statistical analysis plan, and the informed consent form. The data will be made available upon publication to researchers who provide a methodologically sound proposal for use in achieving the goals of the approved proposal. Proposals should be submitted to the corresponding author via email (okafuji@kcho.jp)" All PLOS journals now require all data underlying the findings described in their manuscript to be freely available to other researchers, either 1. In a public repository, 2. Within the manuscript itself, or 3. Uploaded as supplementary information.This policy applies to all data except where public deposition would breach compliance with the protocol approved by your research ethics board. If your data cannot be made publicly available for ethical or legal reasons (e.g., public availability would compromise patient privacy), please explain your reasons on resubmission and your exemption request will be escalated for approval. 3. PLOS requires an ORCID iD for the corresponding author in Editorial Manager on papers submitted after December 6th, 2016. Please ensure that you have an ORCID iD and that it is validated in Editorial Manager. To do this, go to ‘Update my Information’ (in the upper left-hand corner of the main menu), and click on the Fetch/Validate link next to the ORCID field. This will take you to the ORCID site and allow you to create a new iD or authenticate a pre-existing iD in Editorial Manager.

**Additional Editor Comments:**

The abstract should be structured

P-values should be included in the results section of the abstract

In terms of pain assessment, using only the FLACC scale may not provide a comprehensive view

The term "two-group" in the title is unnecessary

Consider adding a row in the tables to compare median scores between guardians and third-party evaluators would improve the strength of the results, especially if no statistical significance is found between the two groups.

The sample size calculation should reference the article it is based on, ensuring proper citation.

The use of a cooling pack may seem outdated and Im not sure about the novelty of this work. Consider discuss it in the introduction first

Different vaccines have varying pain levels, so generalizing across all vaccines is inappropriate.

Tables should not be placed in the introduction

The delay in submitting the study, conducted three years ago, should be explained.

A rationale for including children aged 3–6 years is needed, particularly since not all age groups were included. The age range should also be mentioned in the title.

The inclusion and exclusion criteria need more detail, especially regarding skin diseases, which could impact the effect of the cooling pack.

It’s important to confirm whether the normality of variables was assessed.

When reporting median and IQR, use the format *median [IQR]*

Ensure consistent P-value formatting throughout the manuscript (e.g., *P-value = 0.08* or *P-value < 0.05*).

Each table should include a footer explaining abbreviations and tests used for P-values. The claim that this study is the "first" to investigate local cooling during vaccinations should be reconsidered, as there are many existing studies on this topic.

If the focus is on "childhood anxiety and fear of pain," baseline anxiety should be measured and controlled for, as this can influence the results.

Finally, update the manuscript with recently published studies to reflect the current state of research.

Reviewers' comments:

Reviewer's Responses to Questions

**Comments to the Author**

1. Is the manuscript technically sound, and do the data support the conclusions?

Reviewer #1: Yes

Reviewer #2: Yes

Reviewer #3: Yes

2. Has the statistical analysis been performed appropriately and rigorously? 

Reviewer #1: No

Reviewer #2: Yes

Reviewer #3: Yes

3. Have the authors made all data underlying the findings in their manuscript fully available?

Reviewer #1: Yes

Reviewer #2: Yes

Reviewer #3: Yes

4. Is the manuscript presented in an intelligible fashion and written in standard English?

Reviewer #1: Yes

Reviewer #2: Yes

Reviewer #3: Yes

5. Review Comments to the Author

Reviewer #1: The rationale for not employing the Wilcoxon test for sample size calculation remains unclear.

For assessing reliability, it would be more appropriate to utilize either Spearman's rank correlation coefficient or Cohen's kappa statistic.

Figure 4: the cooling group has less than 15 data points.

Reviewer #2: The article evaluates the effectiveness and safety of using a cooling pack to reduce pain during subcutaneous vaccine injections in children aged 3-6 years. The study was conducted in two pediatric clinics in Japan. Sixty children were randomly assigned to either a cooling group or a non-cooled group. The primary pain assessment checklist was the FLACC scale, rated by a third party who viewed videos of the vaccination. The study found that the cooled group had significantly lower FLACC scores compared to the non-cooled group.

The article discuss a innovative approach to pediatric pain management by introducing a simple, non-pharmacological method using a cooling pack to reduce vaccine injection pain in children but after carefully reviewing the submitted article following points are provided for consideration.

Areas for Consideration:

1) While the components (solutions) of vaccines could differ in various regions, it is important that the brand names of the vaccines used in the study be specified. This is an important omission because different formulations, even for the same type of vaccine, can have varying components or solvents that might influence the level of pain experienced during injection. Additionally, the discussion section mentions that pneumococcal vaccines are known to cause severe pain. Providing this information would be beneficial for future studies to investigate the solvents and other components of the vaccines used in this study as potential contributors to pain. Understanding the role of these components could provide insights into pain management strategies during vaccination.

2) The terminology used in the article should be double-checked and unified. For example, the abstract states, “Local cooling during subcutaneous vaccine administration is a safe and effective pain-relief method and can be implemented in infants receiving subcutaneous vaccines,” while the study actually focuses on children aged 3-6 years. Inconsistency should be corrected to ensure clarity and accuracy throughout the manuscript.

3) The structure of the introduction in the manuscript requires revision to better align with standard scientific practices. The final paragraph of the introduction should focus on clearly stating the objectives and main aim of the study, rather than discussing methodological details or presenting a table. Currently, the introduction ends with a table (Table 1) that details the FLACC scale, which is more appropriate for the Methods section. I recommend that the authors transfer this table to the Methods section and revise the introduction to emphasize the study's objectives more clearly. This adjustment will help maintain the logical flow of the manuscript and ensure that the introduction provides a concise and focused overview of the study's purpose.

4) The FLACC scale, while widely used, is not without its limitations, particularly in a population as young as 3 years old, where subjective pain expression can be highly variable and influenced by factors other than pain itself, such as fear or fatigue. The reliance on a third-party assessment based on video footage further complicates the interpretation, as subtle cues that could indicate discomfort might be missed or misinterpreted. In this case, to enhance the reliability of the results, it would be beneficial to double-check the videos and scoring with another independent assessor. Comparing the scores from multiple assessors could help ensure reproducibility and minimize potential bias or errors in the interpretation of the pain levels observed in the children.

Reviewer #3: In this paper, the author evaluated the pain-relieving effect and safety of local cooling during subcutaneous vaccine injection in young children (3 to 6 years old). This two-group, single-blind, randomized, parallel-group multicenter study demonstrated that a simple and easy method of local cooling could significantly alleviate pain associated with vaccination in young children. It is a well written paper that needs a minor revision and below are my comments:

1. There is strong evidence supporting the effectiveness of topical anesthetics in preventing pain in individuals ≤ 12 years of age. The author did not mention the advantage of current cooling method compared with topical anesthetics or the disadvantage of topical anesthetics in the current treatment in the introduction.

2. In this paper, the author mentioned that PURU CURE Ice Pack® was modified to be small enough to fit the infant’s upper arm. It would be great if the author could compare the price between current Ice pack and the topical anesthetics or spray cooling.

3. Regarding the experiment design, some children may find the cold uncomfortable and thus may show some FLACC behavioral pain after the local cooling. They may show less FLACC behavioral pain for subcutaneous injection in this situation. It would be more convincing if the author could provide the data showing no FLACC difference before and after the local cooling.

4. As author mentioned in the paper, there’re some limitations in the experiment design such as big age differences (3 to 6 ages), 2 different vaccine types and single-blind trial design. There are some other factors that have been demonstrated in reducing the pain need to consider in real situations such as the positioning for needle and the surroundings environment for the children (parents or other people who can distract the child).

5. The pain-relieving effect was tested in Japanese encephalitis virus vaccine or the influenza virus vaccine. In the future study, it would be interesting to see the effects of local cooling on more painful vaccinations such as MMR and HPV.

6. PLOS authors have the option to publish the peer review history of their article (what does this mean? ). If published, this will include your full peer review and any attached files.

**Do you want your identity to be public for this peer review?** For information about this choice, including consent withdrawal, please see our Privacy Policy .

Reviewer #1: No

Reviewer #2: No

Reviewer #3: No

---

## [Author Response · Author response to Decision Letter 1]

10 Nov 2024

Additional Revisions Based on Reviewer and Editor Comments:

• We have now structured the abstract according to journal guidelines.

• P-values have now been included in the results section of the abstract.

• Additional detail on the inclusion/exclusion criteria has been provided, and the age range has been added to the title.

• We have confirmed the normality of variables and clarified the format for median and interquartile range (IQR) reporting.

Additional Comments:

1. Reviewer Comment: “The delay in submitting the study, conducted three years ago, should be explained.”

Response: We appreciate your understanding regarding this delay. Indeed, several factors contributed to the time taken for submission, including our unfamiliarity with the submission process and prolonged response times from previous journals. Furthermore, our clinical workload, which has been exacerbated by staff shortages, limited the time available for writing. Despite these challenges, we remain committed to publishing this study, as we believe it will significantly benefit children by alleviating injection-related pain.

2. Reviewer Comment: “The inclusion and exclusion criteria need more detail, especially regarding skin diseases, which could impact the effect of the cooling pack.”

Response: We considered the potential impact of skin diseases on the cooling pack’s efficacy. However, vaccinations are typically not administered to areas affected by significant skin conditions that may interfere with the cooling pack’s effect. Therefore, we did not find it necessary to explicitly include this in the inclusion/exclusion criteria. If further clarification is required, we will be happy to provide additional details.

---

## [Decision Letter · Decision Letter 1]

26 Nov 2024

PONE-D-24-27274R1Use of a cooling pack to reduce subcutaneous vaccine injection pain in children aged 3-6 years: A single-blind, randomized, parallel-group multicenter studyPLOS ONE

Dear Dr. Okafuji,

Thank you for submitting your manuscript to PLOS ONE. After careful consideration, we feel that it has merit but does not fully meet PLOS ONE’s publication criteria as it currently stands. Therefore, we invite you to submit a revised version of the manuscript that addresses the points raised during the review process.

I appreciate your efforts in addressing the comments. However, as part of the editorial process, it is essential to provide a point-by-point response to the reviewer comments. I kindly request you to:

**Provide a point-by-point response to the comments** in the same order as they were provided.**Indicate the specific parts of the manuscript where changes were made** , with detailed references to line and page numbers.

We look forward to receiving your revised manuscript.

Kind regards,

Seyedeh Yasamin Parvar, M.D., M.P.H.

Academic Editor

PLOS ONE

Journal Requirements:

Additional Editor Comments:

I appreciate your efforts in addressing the comments. However, as part of the editorial process, it is essential to provide a point-by-point response to the reviewer comments. This facilitates a clear understanding of how each point has been addressed.

To ensure clarity and consistency, I kindly request that you:

Provide a point-by-point response to the comments in the same order as they were provided.

Indicate the specific parts of the manuscript where changes were made, with detailed references to line and page numbers.

Additionally, while the manuscript is well-prepared overall, there are a few minor revisions required:

1- Ensure that all typographical issues mentioned previously have been resolved throughout the manuscript.

2- Replace "fig" with "figure" in full and bold this term consistently in the text.

3- In terms of pain assessment, using ONLY the FLACC scale may not provide a comprehensive view.

4- Provide references for this part in the introduction: “A gel-based cooling pack, which maintains its softness even when frozen, may provide a less intimidating alternative for pediatric patients; thus offering a more comfortable and effective pain-relief method during vaccination.”

5- Explain the delay in submitting the study, conducted three years ago, in the “limitations.”

6- Confirm whether the normality of variables was assessed and specify this in detail in the statistical part of the methods section.

7- Rewrite the first sentence of the discussion for clarity.

8- I couldn’t find answer and revisions for this comment. If the focus is on "childhood anxiety and fear of pain," baseline anxiety should be measured and controlled for, as this can influence the results.

9- UPDATE the manuscript with recently published studies to reflect the current state of research.

10-Provide references for this part in the discussion: “While cooling packs are reusable and cost-effective in the long term, they require upfront investment and preparation by hospitals, and the cost may fall on families if purchased individually (3,200 yen per unit).”

11- Avoid using references in the conclusion.

Reviewers' comments:

Reviewer's Responses to Questions

**Comments to the Author**

1. If the authors have adequately addressed your comments raised in a previous round of review and you feel that this manuscript is now acceptable for publication, you may indicate that here to bypass the “Comments to the Author” section, enter your conflict of interest statement in the “Confidential to Editor” section, and submit your "Accept" recommendation.

Reviewer #1: All comments have been addressed

Reviewer #2: All comments have been addressed

Reviewer #3: All comments have been addressed

2. Is the manuscript technically sound, and do the data support the conclusions?

Reviewer #1: (No Response)

Reviewer #2: Yes

Reviewer #3: Yes

3. Has the statistical analysis been performed appropriately and rigorously? 

Reviewer #1: (No Response)

Reviewer #2: Yes

Reviewer #3: Yes

4. Have the authors made all data underlying the findings in their manuscript fully available?

Reviewer #1: (No Response)

Reviewer #2: Yes

Reviewer #3: Yes

5. Is the manuscript presented in an intelligible fashion and written in standard English?

Reviewer #1: (No Response)

Reviewer #2: Yes

Reviewer #3: Yes

6. Review Comments to the Author

Reviewer #1: All my concerns are addressed.

Reviewer #2: (No Response)

Reviewer #3: In this paper, the author demonstrated local cooling could significantly alleviate pain associated with subcutaneous vaccine injection in young children (3 to 6 years old). Compared with topical anesthetics, local cooling offers rapid pain relief within 30 to 60 seconds, making it a more practical option for pediatric use. Although FLACC scale has the limitation for young children, this gel-based cooling pack still demonstrated the effectiveness of local cooling in pain-relief in clinical settings, which is of great importance. I would recommend this paper to be published in PLOS ONE journal.

7. PLOS authors have the option to publish the peer review history of their article (what does this mean? ). If published, this will include your full peer review and any attached files.

**Do you want your identity to be public for this peer review?** For information about this choice, including consent withdrawal, please see our Privacy Policy .

Reviewer #1: No

Reviewer #2: No

Reviewer #3: No

---

## [Author Response · Author response to Decision Letter 2]

29 Dec 2024

Point-by-Point Responses

General Editorial Comments

Comment: Provide a point-by-point response to the comments in the same order as they were provided. Indicate the specific parts of the manuscript where changes were made, with detailed references to line and page numbers.

Response: We have followed this guideline throughout the rebuttal letter. Each response is organized sequentially, and includes specific references to the revised manuscript.

Minor Revisions

1. Typographical Issues

Comment: Ensure that all typographical issues mentioned previously have been resolved throughout the manuscript.

Response: We have reviewed the entire manuscript to ensure that all such typographical errors have been resolved.

2. Use of "Figure"

Comment: Replace "fig" with "figure" in full and bold this term consistently in the text.

Response: All instances of "fig" have been replaced with "figure" throughout the manuscript. Examples can be seen on Page 6, Line 108 and 117

Specific Revisions Requested by the Editor

3. FLACC Scale

Comment: Using only the FLACC scale may not provide a comprehensive view.

Response: We agree that using only the FLACC scale may not provide a sufficiently comprehensive view. While the FLACC scale was chosen for its suitability for assessing pain in young children, we recognize its limitations in capturing subjective patient experiences such as fear and anxiety. We have added a discussion of this limitation and the potential benefits of incorporating other tools, such as the Children's Fear Scale, in future studies [Page 17, Line 325-329]

4. Cooling Pack Reference

Comment: Provide references for this part in the introduction: "A gel-based cooling pack, which maintains its softness even when frozen, may provide a less intimidating alternative for pediatric patients; thus offering a more comfortable and effective pain-relief method during vaccination."

Response: We acknowledge that the original statement lacked supporting references. To address this, we have revised the text to clarify that the described benefits of gel-based cooling packs are based largely on anecdotal observations in clinical practice rather than systematic evidence. Additionally, we have noted the requirements for further studies to substantiate this impression. [Page 5, Line 80-84]

5. Delay in Submission

Comment: Explain the delay in submitting the study, conducted three years ago, in the "limitations."

Response: In response to your comment, we have added a brief explanation to the "Limitations" section clarifying that logistical challenges and prioritization of clinical duties contributed to this delay. [Page 17, Lines 329-332]

Statistical and Methodological Revisions

6. Normality of Variables

Comment: Confirm whether the normality of variables was assessed and specify this in detail in the statistical part of the methods section.

Response: We decided in advance in the protocol that we would conduct a non-parametric test because we thought that the distribution of FLACC scores would not follow a normal distribution, and we did not conduct a non-parametric test after examining the normality of the data obtained, retrospectively.

Discussion Section Revisions

7. First Sentence of the Discussion

Comment: Rewrite the first sentence of the discussion for clarity.

Response: The first sentence of the Discussion has been rewritten as follows: "This is the first study to reveal that applying local cooling prior to vaccination can alleviate pain associated with vaccination in young children aged 3–6 years." (Page 15, Line 272)

8. Baseline Anxiety

Comment: I couldn’t find answer and revisions for this comment. If the focus is on "childhood anxiety and fear of pain," baseline anxiety should be measured and controlled for, as this can influence the results.

Response: We acknowledge that baseline anxiety can significantly impact a child's perception of pain during vaccination. In the present study, we employed randomization to minimize the impact of baseline differences, including anxiety, between the groups. While we did not specifically measure baseline anxiety, randomization ensured that potential differences in baseline anxiety are distributed evenly across the groups, thereby reducing their influence on outcomes. We have added this limitation and suggested that future studies incorporate baseline anxiety measurements. [Page 17, Line 321-325]

9. Updated Literature

Comment: Update the manuscript with recently published studies to reflect the current state of research.

Response: We have included references to four recently published studies (References 12, 13, 26, and 27) to better reflect the current state of research.

10. Cost Reference for Cooling Packs

Comment: Provide references for this part in the discussion: "While cooling packs are reusable and cost-effective in the long term, they require upfront investment and preparation by hospitals, and the cost may fall on families if purchased individually (3,200 yen per unit)."

Response: We acknowledge that the statement in question lacked supporting references. To address this, we have clarified that the cost-effectiveness of cooling packs has not been rigorously investigated, and that the statement reflects general observations rather than specific research findings. [Page 16, Line 307- Page17, L311]

Conclusion Section

11. Avoiding References

Comment: Avoid using references in the conclusion.

Response: We have removed the reference from the conclusion section and rephrased the text to summarize the key findings concisely. [Page 18, Lines 338-342]

---

## [Decision Letter · Decision Letter 2]

15 Jan 2025

Use of a cooling pack to reduce subcutaneous vaccine injection pain in children aged 3-6 years: A single-blind, randomized, parallel-group multicenter study

PONE-D-24-27274R2

Dear Dr. Okafuji,

We’re pleased to inform you that your manuscript has been judged scientifically suitable for publication and will be formally accepted for publication once it meets all outstanding technical requirements.

Kind regards,

Seyedeh Yasamin Parvar, M.D., M.P.H.

Academic Editor

PLOS ONE

Additional Editor Comments (optional):

Reviewers' comments:

Reviewer's Responses to Questions

**Comments to the Author**

1. If the authors have adequately addressed your comments raised in a previous round of review and you feel that this manuscript is now acceptable for publication, you may indicate that here to bypass the “Comments to the Author” section, enter your conflict of interest statement in the “Confidential to Editor” section, and submit your "Accept" recommendation.

Reviewer #1: All comments have been addressed

2. Is the manuscript technically sound, and do the data support the conclusions?

Reviewer #1: (No Response)

3. Has the statistical analysis been performed appropriately and rigorously? 

Reviewer #1: (No Response)

4. Have the authors made all data underlying the findings in their manuscript fully available?

Reviewer #1: (No Response)

5. Is the manuscript presented in an intelligible fashion and written in standard English?

Reviewer #1: (No Response)

6. Review Comments to the Author

Reviewer #1: All my comments have been addressed.

7. PLOS authors have the option to publish the peer review history of their article (what does this mean? ). If published, this will include your full peer review and any attached files.

**Do you want your identity to be public for this peer review?** For information about this choice, including consent withdrawal, please see our Privacy Policy .

Reviewer #1: No

---

## [Editor Report · Acceptance letter]

PONE-D-24-27274R2

PLOS ONE

Dear Dr. Okafuji,

I'm pleased to inform you that your manuscript has been deemed suitable for publication in PLOS ONE. Congratulations! Your manuscript is now being handed over to our production team.

Kind regards,

on behalf of

Dr. Seyedeh Yasamin Parvar

Academic Editor

PLOS ONE